# A Randomized Clinical Trial Investigating an Integrated Nursing Educational Program to Mitigate Chemotherapy-Induced Nausea and Vomiting in Cancer Patients: The NIV-EC Trial

**DOI:** 10.3390/cancers15215174

**Published:** 2023-10-27

**Authors:** Cristina Mazzega-Fabbro, Jerry Polesel, Lara Brusutti, Elisa Malnis, Chiara Sirelli, Annalisa Drigo, Marina Manicone, Monica Rizzetto, Camilla Lisanti, Fabio Puglisi

**Affiliations:** 1Nursing Team, Department of Medical Oncology, Centro di Riferimento Oncologico di Aviano (CRO) IRCCS, 33081 Aviano, Italy; lara.brusutti@cro.it (L.B.); elisa.malnis@cro.it (E.M.); chiara.sirelli@cro.it (C.S.); annalisa.drigo@cro.it (A.D.); mmanicone@cro.it (M.M.); 2Department of Medical Sciences, Nursing School, University of Udine, 33100 Udine, Italy; 3Unit of Cancer Epidemiology, Centro di Riferimento Oncologico di Aviano (CRO) IRCCS, 33081 Aviano, Italy; polesel@cro.it; 4Department of Medical Oncology, Centro di Riferimento Oncologico di Aviano (CRO) IRCCS, 33081 Aviano, Italy; monica.rizzetto@cro.it (M.R.); camilla.lisanti@cro.it (C.L.); fabio.puglisi@cro.it (F.P.); 5Department of Medicine, University of Udine, 33100 Udine, Italy

**Keywords:** nausea, vomiting, patient education as topic, clinical nursing research

## Abstract

**Simple Summary:**

Chemotherapy-induced nausea and vomiting is a prevalent and distressing side effect of chemotherapy, greatly impacting patient quality of life. In addition to pharmacological prevention, nausea and vomiting can be mitigated through patient education on their prevention and management, especially when supporting material is provided. The present randomized controlled trial evaluated the use of an informative booklet with details on antiemetic drugs and nutritional recommendations to reduce CINV provided to patients in addition to oral information. Cancer patients undergoing first chemotherapy cycle were randomly assigned to receive oral information regarding CINV prevention and management (control arm) or oral information plus an informative booklet (experimental arm). Patients in the experimental arm reported a lower frequency of nausea occurrence in the first five days after chemotherapy by about 8%, compared to those in the control group. Although the beneficial effect was moderate, this intervention demands minimal resources in terms of costs and time.

**Abstract:**

Background: In addition to pharmacological prevention, chemotherapy-induced nausea and vomiting (CINV) can be mitigated through patient education; written supporting materials can be beneficial. Methods: This is a randomized, controlled trial which randomly assigned patients undergoing first chemotherapy cycle to receive oral information regarding CINV prevention and management (control arm) or oral information plus an informative booklet (experimental arm). Overall, 384 cancer patients fulfilling the following inclusion criteria were enrolled: age ≥18 years; life expectancy ≥6 months; no cognitive impairment; written informed consent. After the first cycle, CINV occurrence and its impact on daily activities were assessed using the Functional Living Index Emesis (FLIE). Results: Severe nausea was self-reported by 3.0% and 10.8% of patients in the experimental and control group, respectively (difference: 7.8%; 95% confidence interval: 2.3% to 13.1%). Moderate/high impact of nausea on daily activities was lower in patients also receiving the booklet than in the control group (4.2% and 10.1%, respectively; difference: 5.9%; 95% confidence interval: 0.3% to 11.5%). Vomiting was not statistically different between study arms. Conclusions: This integrated nursing approach was effective in aiding cancer patients in CINV self-management. Although the beneficial effect was moderate, this intervention demands minimal resources in terms of costs and time.

## 1. Introduction

Chemotherapy-induced nausea and vomiting (CINV) is a frequent toxicity in patients undergoing oncological treatment [1,2,3]; despite the advances in pharmacological prevention of CINV, vomiting occurs in 40% of patients treated with high-to-medium emetogenic drugs, whereas nausea affects 70−80% of patients undergoing chemotherapy [4]. Nausea and vomiting are two basic protective reflexes against the absorption of toxins and in response to well-defined signals. Chemotherapy can trigger vomiting by acting directly at the level of chemoreceptor trigger zone (CTZ) or by indirect action on the gastrointestinal tract [2]. CINV are classified into five categories according to the timing of onset: anticipatory (when it occurs before chemotherapy administration in patients who have already experienced acute or delayed emesis), acute (when it occurs within 24 h of receiving chemotherapy), delayed (when it occurs more than 24 h after receiving chemotherapy), breakthrough (when it occurs within five days of chemotherapy despite appropriate prophylaxis) and refractory (when it occurs in subsequent chemotherapy cycles despite appropriate prophylaxis) [5]. The severity and frequency of CINV depend on several factors, such as the emetogenic potential of the chemotherapy regimen, the dosage of administered drugs, the therapeutic program, the route of administration, the type and stage of tumor, some patient’s characteristics (i.e., female sex, young age), comorbidities, and concomitant medications [4,6,7].

CINV leads to the appearance of possible complications, such as electrolyte and metabolic disorders, dehydration and weight loss up to malnutrition and anorexia [8]. The scientific literature showed that vomiting related to chemotherapy is significantly decreased after the introduction in clinical practice of serotonin receptor antagonists, while control of nausea is more challenging, especially in the first 24 h after administration of the therapy [9]. During oncological treatments, antiemetic therapy should be re-evaluated and adapted over time taking into account patient clinical conditions and possible changes into the therapeutic process [10]. The American Society of Clinical Oncology (ASCO), the National Comprehensive Cancer Network (NCCN), the Multinational Association of Supportive Care in Cancer (MASCC) and the European Society of Medical Oncology (ESMO) have developed some recommendations for the management of CINV that, when applied, increase its control by 20% [11]. In addition, the probability of not developing vomiting increases by up to 43% in patients who also receive guideline-compliant prophylaxis [11]. Over the years, many studies have focused on the identification and comparison of the various antiemetic drugs available and their possible combinations, without dwelling on the importance of therapeutic education and personalized patient information [9].

Therefore, prevention is essential to avoid the appearance of anticipated or uncontrolled nausea and vomiting induced by chemotherapy. This may also help with prevention and management of psychological factors and anxiety associated with CINV [12]. Some studies showed that the CINV can be significantly reduced with educational support [13,14,15,16,17,18]. Nurses may play a crucial role in this educational process [16,17]. However, up to date, validated written tools are scarce and rarely used, so that information on the patients’ prevention and self-management of CINV is usually mainly given by oral form [16,19]. To increase the patient’s knowledge about CINV and to improve self-management of antiemetic drugs at home, information should be given through both verbal and written materials [4].

As far as we know, no study has been conducted in Italy with the focus on investigating educational tools or methods capable of reducing CINV through the empowerment of cancer patients. Since the cultural background and language are relevant mediators of the educational process, the aim of our study was to evaluate the role of a nursing integrated educational approach (i.e., an informative booklet plus an oral educational session) in reducing delayed and breakthrough CINV in Italian cancer patients undergoing to a first-line chemotherapy.

## 2. Materials and Methods

A randomized controlled trial was conducted to evaluate the efficacy of oral plus written information versus oral information alone in reducing the incidence of CINV of moderate-to-severe intensity in patients undergoing first-line chemotherapy. The study focused on delayed and breakthrough CINV, i.e., that occurring from 24 h to five days after chemotherapy administration [9,11].

### 2.1. Patients

The present study was conducted at the Multidisciplinary Day Hospital of the Oncology Department of the Centro di Riferimento Oncologico di Aviano (CRO), in the north of Italy. Between December 2018 and November 2022, 384 cancer patients were enrolled; recruitment was suspended for nine months between March and November 2020 because restrictions in hospital admission due to concomitant SARS-CoV-2 pandemic. Patients were enrolled in the study if they met the following inclusion criteria: (a) age ≥18 years; (b) undergoing first cycle of first-line chemotherapy; (c) capability to fill-in the questionnaires; (d) signed a written informed consent. Exclusion criteria included: (a) non-cooperative patients; (b) patients in end-stage disease (i.e., life expectancy <6 months); (c) psychiatric or neurological comorbidities determining cognitive impairment (e.g., Alzheimer’s and Parkinson’s diseases, senile dementia); (d) low vision. Before enrollment, patients were informed about study aims and procedures, reporting that the participation was entirely on a voluntary base, and that they have the right to withdraw from the study at any time. All participants signed a written specific informed consent.

### 2.2. Study Procedures

Eligible patients were randomized (ratio: 1:1) through hidden allocation to one of the two study arms (Figure 1): in the control arm, patients received a 30-min oral educational session about CINV, while in the experimental arm, they also received an informative booklet about CINV prevention and self-management strategies. The 30-min educational oral session was performed in both study arms before the administration of the first cycle of first-line chemotherapy; it was carried out by an instructed nurse who illustrated the main characteristics of CINV, the common strategies to prevent and manage it, the common antiemetic drugs that patients could take at home and some nutritional recommendations. The experimental arm further received an informative booklet including two sections: (a) a list of antiemetic drugs used at home and for CINV self-medication, with specific dosages and timing of intake; (b) practical dietary recommendations which have been shown to be useful in reducing the occurrence of CINV, such as: eating small and frequent meals during the day, preferring simple and dry foods and avoiding spicy and acidic foods.

At enrollment, the nursing staff retrieved socio-demographic and clinical data from the medical records, including age, sex, cancer diagnosis, treatment programs, and antiemetic drugs already planned as standard prophylaxis (according to chemotherapy regimen). Information about occurrence, severity and frequency of CINV were assessed after the first cycle of chemotherapy through the Functional Living Index Emesis (FLIE) [20], a validated patient-reported outcome instrument. The FLIE comprised two domains (vomiting and nausea) with nine identical items in each. The first item in each domain asked the patients to rate how much nausea (vomiting) they have experienced over the past five days. The remaining eight items assessed the impact of CINV on the following aspects of patient’s daily life: ability to enjoy meals/liquids, ability to prepare meals/do household tasks, ability to perform daily functions, ability to perform usual recreation/leisure activities, willingness to spend time with family and friends, extent to which the side effects have caused personal hardship and hardship on others. Each item was answered using a 100 mm (1 to 7 points) visual analog scale (VAS) with anchors corresponding to “None”/”Not at all” and “A great deal”, and tick-marks dividing the scale into six equal categories. For each item, the answer was categorized into three levels: “None/Not at all” for item score ≤ 2; “A little/moderately” for item score 3–4; “A lot/A great deal” for item score ≥ 5. Total score was further calculated overall and for nausea and vomiting subscales by summing up the responses of the relevant items. The total score was then converted into a percentage by dividing by the total, with 0% and 100% representing the lowest and highest levels of impact of nausea and vomiting on daily activities. Total scores were finally categorized in three levels: absent (0%), mild (1% to 49%), and moderate-to-high (≥50%).In addition, anxiety state was assessed before the first and second chemotherapy cycles through the Hamilton Anxiety Rating Scale (HAM-A) [21]. It consisted of 14 symptom-defined elements and catered for both psychological and somatic symptoms. Each item was scored on a basic numeric scoring of 0 (not present) to 4 (severe) and summed up to total. Four levels of anxiety were defined on the basis of the total HAM-A score: absent (score ≤ 7), mild (8–17), moderate (18–24), and severe (≥25).

### 2.3. Statistical Analyses

The study required enrollment of 163 patients for each arm to evaluate a difference of 15% in the occurrence of nausea (i.e., 45% in patients receiving oral information versus 30% in those receiving oral and written information) with a priori error probabilities of α = 0.05 and β = 0.20. Expecting a dropout of 15% due to patients lost at second chemotherapy administration or incomplete data, a total of 384 patients had to be enrolled.

Distributions of categorical variables across study arms were compared through Fisher’s exact test. The prevalence of nausea and vomiting was reported as percentage with corresponding 95% confidence interval according to the Clopper–Pearson method. Prevalence of nausea and vomiting across study arms were compared through Pearson’s χ^2^ test. Further, the relative risk (RR) for not developing an adverse outcome (e.g., severe nausea) in the experimental arm compared to controls—with corresponding 95% confidence interval (CI)—was calculated. Statistical significance was claimed for *p* < 0.05 (two-sided).

## 3. Results

Overall, 435 patients were evaluated for eligibility: 49 did not meet the inclusion criteria and two refused participation, thus leaving 384 patients for randomization (Figure 1). Fifty-eight patients (15.1%) were excluded for incomplete data in both nausea and vomiting FLIE subscales. 

The socio-demographic and clinical characteristics of patients were similar across each study arm (Table 1). The majority of enrolled patients were women aged ≥50 years; breast cancer was the most frequent cancer type in both groups. Chemotherapy regimens were similar in both study arms, with epirubicin + cyclophosphamide being the most frequent chemotherapy scheme; similarly, the use of potentiated antiemetic premedication—including aprepitant, increased corticosteroids dose, and benzodiazepines—was received by more than 50% of patients in each arm. Moderate-to-severe anxiety (according to the total HAM-A score) at first chemotherapy administration was also similar in the two study groups (9.0% and 11.9%, *p* = 0.748), as it was before the second administration (7.8% and 8.1%, *p* = 0.915).

After the first chemotherapy cycle, severe nausea—defined as a score ≥5 in the specific FLIE item—was self-reported by five patients (3.0%) in the experimental arm and in 17 patients (10.8%) in the control arm (Table 2), with a statistically significant reduction of incidence of 7.8% (95% CI: −13.1% to −2.3%; *p* = 0.006); consequently, the risk of occurrence of severe nausea was reduced by 70% (RR = 0.28; 95% CI: 0.11–0.74). Overall, moderate-to-high impact of nausea on daily activities was lower in the experimental arm than in the control arm (4.2% and 10.1%, respectively; difference: −5.9%; 95% CI: −11.5% to −0.3%; *p* = 0.038). The proposed integrated educational program significantly lowered the impairment of patients in pursuing leisure activities (2.4% in the experimental arm and 8.2% in the control arm; difference: −5.8%; 95% CI: −10.7% to −0.9%; *p* = 0.019), with a reduction of 70% of the risk that nausea limited such activities (RR = 0.29; 95% CI: 0.10–0.88). Results about all nausea items are reported in Appendix A. Conversely, the difference between study arms in vomiting prevention was not statistically significant, with 2.6% and 2.1% of patients reported severe vomiting in the experimental and control group, respectively (*p* = 0.771).

The reduction of the impact of the moderate-to-high nausea on daily activities was independent of sex, age, and use of prophylactic antiemetic drugs (Table 3). Patients with baseline moderate-to-severe depression reported higher impact of nausea in both arms (13.3% and 15.8%, respectively). Interestingly, the reduction effectiveness of the integrated approach in reducing the impact of nausea on daily activities increased with decreasing level of depression, rising from −2.5% (95% CI: −26.2% to 21.3%) in patients with moderate-to-severe depression to −8.6% (95% CI: −17.7% to 0.6%) in those with no depression. Differences were, however, not statistically significant due to low number of patients in the strata.

## 4. Discussion

Our study shows that an integrated educational program reduces the incidence of severe nausea in the five days after the first cycle of chemotherapy by about 8% compared to oral information alone. Patients in the experimental arm further report statistically significant lower impact of nausea on their daily activities than in the control arm. Interestingly, the efficacy of this intervention is inversely associated with the baseline level of patient depression. However, our integrated educational intervention seems to not have a significant impact on vomiting.

Despite the developments in prophylactic drugs, CINV continues to be the most frequent side effect in patients undergoing chemotherapy treatment [4], with great impact on patient quality of life and treatment adherence. Although several supportive care strategies have been proposed [22,23], patient empowerment remains a crucial point to improve the adherence to CINV prophylactic strategies and the prevention of general chemotherapy side effects [24,25]. Indeed, most of the antiemetic drugs are self-administered at home so treatment adherence and CINV tolerance could be increased by instructing patients on antiemetic drug assumptions and on recognizing antiemetic regimens’ adverse events [26]. Furthermore, nutritional counseling could help in lessening nausea and vomiting. In addition to oral information, an educational program should also include written materials appropriate to the health literacy level of the patients [4].

Our findings on nausea are in agreement with the results of previous investigations aiming to provide patients with written support for the management of CINV. Three controlled studies conducted in Turkey [13,14,16] evaluated the efficacy of a three-session educational program using a literature-based booklet in reducing CINV occurring in the first five days after the first cycle of chemotherapy. These studies reported a significant reduction in the frequency and intensity of nausea in the experimental group in comparison to standard oral education, as well as a reduction in the occurrence of other gastroenteric adverse events such as diarrhea. The highest reduction was seen in the first 24 h following chemotherapy administration, and the gap remained stable up to five days thereafter [16]. Similarly, an Iranian controlled trial [15] reported a reduction in the intensity of CINV in patients receiving a 12-session educational intervention with pamphlets, brochures and videos. One additional study [17] evaluated the use of a mobile phone software to provide patients with recommendations and dietary advice for CINV reduction after chemotherapy administration: severity of nausea was reduced, but not the proportion of affected patients. To be noted as in all these studies, the educational program was organized in three or more sessions and required significant economic and time resources. Only an Australian study [27] proposed a single educational session prior to chemotherapy administration, using information sheets on antiemetics and self-care; unfortunately, this study did not reported a significant reduction in nausea occurrence after chemotherapy. In regard to vomiting, a positive effect in reducing its frequency and severity has been generally reported after educational intervention with written information [13,14,15], though some studies did not find any effect [16,27].

Although in the present study, the administration of a written booklet had no impact on vomiting, the intervention has a positive effect on patient quality of life, by reducing the impact of nausea on their daily activities. To the best of our knowledge, no previous studies evaluated this aspect through a validated tool (e.g., the FLIE questionnaire), though a reduction of feeling unusual fatigue, feeling weak and difficult sleeping has been associated with the use of support materials [13,14]. Furthermore, a reduction of psychological distress has been reported [13,14,27].

Since CINV often occurs at home, clear instructions should be given to patients during oral educational sessions and take-home material should be provided [26]. In particular, written information—provided as booklet, pamphlet, or mobile-phone app—are crucial for patient empowerment: patients could recover information lost during the oral educational session due to anxiety and emotional state that reduce their learning and attention.

Within a multidisciplinary management considering pharmacological and non-pharmacological approaches, nurses may play a key role in patient education for prevention and management of CINV [13,28,29]. An international survey conducted in 2014 [19] revealed that over two-thirds of nurses working in oncological wards were constantly involved in CINV assessment, and almost all nurses reported requests of information by patients about antiemetic prophylaxis and drugs used. Similar findings were reported by a previous study in Hong Kong [30], which further highlighted that only 76% of nurses contributed to patient education for CINV management, despite there being a massive request for information by the patients. However, this study reported that information was often provided without a planned educational program and without the use of any supporting materials. Lack of specific training and heavy workload were the major causes preventing nurses from delivering adequate patient education [19,30]. Indeed, 43% of surveyed nurses perceived to have inadequate knowledge about CINV risk assessment and prevention [19]. Improving nurses’ communication skills would also increase the efficacy of educational interventions [31]. A recent meta-analysis reported that the nurses undergoing communication training were less likely to use unnecessary content during conversations with patients and were more likely to use open and evaluative questioning styles [31]. Interestingly, these may have direct effects on patients’ quality of life: a randomized controlled study conducted in cancer patients reported a more positive emotional state in those engaged with nurses who attended a three-day communication course [32], though anxiety level was not significantly reduced.

Some limitations have to be acknowledged. Although the sample size was the largest among similar studies [13,14,15,16,17], subgroup analyses were nonetheless limited by the low frequency of moderate-to-high nausea. Therefore, results about the interaction with sex, age, antiemetic prophylactic drug use, and anxiety state at baseline should be considered as explorative. Further, we noted that the duration of the oral education intervention was significantly longer in the control arm than in the experimental arm. Although it may have negatively impacted the efficacy of the integrated approach, it was considered unethical to refuse a specific patient’s requests on treatment and nutritional recommendations.

## 5. Conclusions

The present study highlighted that the use of a nursing integrated educational approach, including an informative booklet plus an oral session, was more efficient in reducing chemotherapy-associated nausea in cancer patients. Although the beneficial effect affects only 8% of patients, it is worth noting that this nursing intervention is low resource-consuming in terms of cost and time. Therefore, in addition adequate nurse training, including communication skills, an informative booklet should be considered for patient empowerment in order to improve their ability to prevent and manage CINV.

## Figures and Tables

**Figure 1 cancers-15-05174-f001:**
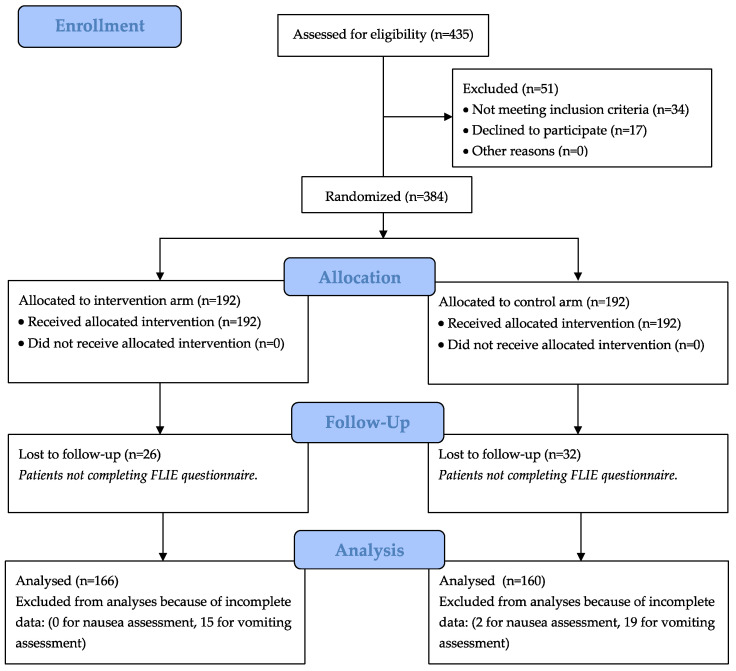
CONSORT flow diagram.

**Table 1 cancers-15-05174-t001:** Sociodemographic and clinical characteristics.

	Study Arm (Type of Information)				Fisher’s Exact Test
	Oral and Written (*n* = 166)		Only Oral (*n* = 160)		
	*n*	(%)	*n*	(%)	
Sex					
Man	32	(19.3)	35	(21.9)	*p* = 0.586
Woman	134	(80.7)	125	(78.1)	
Age (years)					
<50	50	(30.1)	58	(36.3)	*p* = 0.310
50–64	81	(48.8)	65	(40.6)	
≥65	35	(21.1)	34	(23.1)	
Cancer type					
Breast	75	(45.2)	87	(54.4)	*p* = 0.339
Colorectum	29	(17.5)	17	(10.6)	
Ovary	21	(12.7)	17	(10.6)	
Lung	11	(6.6)	15	(9.4)	
Endometrium	9	(5.4)	8	(5.0)	
Other ^1^	21	(12.7)	16	(10.0)	
Chemotherapy type					
Epirubicin + cyclophosphamide	68	(41.0)	69	(43.1)	*p* = 0.317
Paclitaxel + carboplatin	33	(19.9)	28	(17.5)	
XELOX/GEMOX	21	(12.7)	10	(6.3)	
Cisplatin	11	(6.6)	17	(10.6)	
5-FU-based chemotherapy ^2^	14	(8.4)	13	(8.1)	
Other	19	(11.5)	23	(14.4)	
Antiemetic treatment					
Standard	77	(46.4)	69	(43.1)	*p* = 0.491
Potentiated	86	(51.8)	90	(56.3)	
Not reported	3	(1.8)	1	(0.6)	
Anxiety at 1st administration (HAM-A)					
Absent (≤7)	65	(39.2)	61	(38.1)	*p* = 0.748
Mild (8–17)	86	(51.8)	80	(50.0)	
Moderate (18–24)	11	(6.6)	16	(10.0)	
Severe (≥25)	4	(2.4)	3	(1.9)	
Anxiety at 2nd administration (HAM-A)					
Absent (≤7)	75	(45.2)	78	(48.8)	*p* = 0.915
Mild (8–17)	78	(47.0)	69	(43.1)	
Moderate (18–24)	11	(6.6)	11	(6.9)	
Severe (≥25)	2	(1.2)	2	(1.3)	

^1^ Including three patients with multiple cancers. ^2^ Including FOLFOX, FOLFOXIRI, FOLFIRI, and FOLFIRINOX. HAM-A: Hamilton Anxiety Rating Scale.

**Table 2 cancers-15-05174-t002:** Self-reported total nausea and vomiting score (FLIE subscales) according to study arm.

	Study Arm (Type of Information)				Difference % (95% CI)	Pearson’s χ^2^ Test
	Oral and Written (*n* = 166)		Only Oral (*n* = 160)			
	*n*	(%)	*n*	(%)		
Nausea						
Did you experience nausea in the past five days?						
None/Not at all	143	(86.1)	123	(77.9)	8.2 (−0.1 to 16.6)	*p* = 0.052
A little/Moderately	18	(10.8)	18	(11.4)	−1.4 (−7.4 to 6.3)	*p* = 0.875
A lot/A great deal	5	(3.0)	17	(10.8)	−7.8 (−13.2 to −2.3)	*p* = 0.006
Did the nausea affect your ability to pursue leisure activities?						
None/Not at all	142	(85.5)	130	(82.3)	3.3 (−4.7 to 11.3)	*p* = 0.424
A little/Moderately	20	(12.1)	15	(9.5)	2.6 (−4.2 to 9.3)	*p* = 0.459
A lot/A great deal	4	(2.4)	13	(8.2)	−5.8 (−10.7 to −0.9)	*p* = 0.019
Total nausea FLIE subscale						
None (0%)	58	(34.9)	53	(33.5)	1.4 (−8.9 to 11.7)	*p* = 0.791
Mild (1%–49%)	101	(60.8)	89	(56.3)	4.5 (−6.2 to 15.2)	*p* = 0.410
Moderate/high (≥50%)	7	(4.2)	16	(10.1)	−5.9 (−11.5 to −0.3)	*p* = 0.038
Vomiting						
Did you vomit in the past five days?						
None/Not at all	138	(91.4)	124	(87.9)	3.5 (−3.5 to 10.4)	*p* = 0.332
A little/Moderately	9	(6.0)	14	(9.9)	−3.9 (−10.2 to 2.2)	*p* = 0.208
A lot/A great deal	4	(2.6)	3	(2.1)	0.5 (−3.0 to 4.0)	*p* = 0.771
Did vomiting affect your ability to pursue leisure activities?						
None/Not at all	131	(86.8)	123	(87.2)	−0.4 (−8.2 to 7.2)	*p* = 0.903
A little/Moderately	17	(11.3)	16	(11.4)	−0.2 (−7.4 to 7.2)	*p* = 0.981
A lot/A great deal	3	(2.0)	2	(1.4)	0.6 (−2.4 to 3.5)	*p* = 0.708
Total vomiting FLIE subscale						
None (0%)	73	(48.3)	76	(53.9)	−5.6 (−17.0 to 5.9)	*p* = 0.343
Mild (1%–49%)	75	(49.7)	63	(44.7)	5.0 (−6.5 to 16.4)	*p* = 0.394
Moderate/high (≥50%)	3	(2.0)	2	(1.4)		

CI: Confidence interval; FLIE: Functional Living Index Emesis.

**Table 3 cancers-15-05174-t003:** Percentage of patients self-reporting moderate-to-high nausea score (FLIE subscales) according to study arm and patient’s characteristics.

	Study Arm (Type of Information)		Difference % (95% CI)	Pearson’s χ^2^ Test
	Oral and Written (*n* = 166)	Only Oral (*n* = 160)		
Overall	4.2%	10.1%	−5.9 (−11.5 to −0.3)	*p* = 0.038
Sex				
Man	3.1%	8.8%	−5.7 (−17.0 to 5.6)	*p* = 0.332
Woman	4.5%	10.5%	−6.0 (−12.4 to 0.4)	*p* = 0.065
Age (years)				
<50	4.0%	10.3%	−6.4 (−15.9 to 3.2)	*p* = 0.209
50–64	4.9%	9.2%	−4.3 (−12.8 to 4.2)	*p* = 0.308
≥65	2.9%	11.4%	−8.5 (−20.5 to 3.3)	*p* = 0.164
Antiemetic treatment				
Standard	2.5%	9.0%	−6.5 (−14.1 to 1.3)	*p* = 0.097
Potentiated	5.85	11.1%	−5.3 (−13.5 to 2.9)	*p* = 0.208
Anxiety at first infusion (Hamilton scale)				
Absent	3.1%	11.7%	−8.6 (−17.7 to 0.6)	*p* = 0.063
Mild	3.5%	7.6%	−4.1 (−11.1 to 2.9)	*p* = 0.246
Moderate-to-severe	13.3%	15.8%	−2.5 (−26.2 to 21.3)	*p* = 0.841

CI: Confidence interval; FLIE: Functional Living Index Emesis.

## Data Availability

The data are available for research purposes on reasonable request to the corresponding author.

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
