# Peer review of "A Randomized Clinical Trial Investigating an Integrated Nursing Educational Program to Mitigate Chemotherapy-Induced Nausea and Vomiting in Cancer Patients: The NIV-EC Trial"

_cancers, 2023, doi:10.3390/cancers15215174_

Round 1

Reviewer 1 Report

Comments and Suggestions for Authors

Dear authors,

I have found your study very interesting, as whereas little research is done to the specific topic.

It actually focuses on  very common symptoms, chemotherapy- induced in cancer patients. Hopefully, this Integrated Nursing Educational Program, will add to the subject area of patient education in the severe symptoms of nausea and vomiting. Methodology described is fine and the study population is adequate.

Writing style is broadly fine, even though there are very few sentences which should be corrected for the English language. Licensure number and approval is omitted. Please, it is considerable to add it.

Comments on the Quality of English Language

The use of English is very good. 

Reviewer 2 Report

Comments and Suggestions for Authors Congratulate the authors for their work, simply explain that they were prevented from publishing it earlier due to the dates of data collection and the date of receipt of the work.

Reviewer 3 Report

Comments and Suggestions for Authors

This study explored non-pharmacological methods to manage CINV using patient education. In a randomized trial involving 384 patients with cancer, one group received oral CINV prevention information, while another also got an additional informative booklet. Post-chemotherapy, severe nausea was reported by 3.0% in the booklet group, compared to 10.8% without it. The booklet also reduced the nausea's impact on daily activities. However, vomiting rates were unchanged. Conclusively, the educational approach, particularly with the booklet, moderately but cost-effectively aided patients in managing CINV.

- no information about the trial registration?

- It's noteworthy that while nausea outcomes showed differences, vomiting did not. This raises questions about the actual clinical significance of the educational intervention.

- Though the results are statistically significant, the actual impact is moderate, warranting consideration of whether the intervention's benefits substantially outweigh its costs in varied healthcare settings.

- CINV requires a multidisciplinary approach considering pharmacological and non-pharmacological, and there is no point in focusing on patient education alone. 

Reviewer 4 Report

Comments and Suggestions for Authors

Congratulations to your interesting paper! This study represents a significant leap forward in the care and support of cancer patients undergoing chemotherapy. The integrated nursing approach, coupled with the distribution of informative booklets, has demonstrated its effectiveness in reducing severe nausea and minimizing its impact on daily activities. While the benefits may be described as moderate, they are nothing short of life-changing for those undergoing the rigors of cancer treatment. This approach, with its minimal resource requirements, offers a ray of hope for improving the quality of life for cancer patients. It is a testimony to the power of education and support in the face of one of the most challenging aspects of cancer treatment.

Reviewer 5 Report

Comments and Suggestions for Authors

Dear authors and editor,

The manuscript titled "A Randomized Clinical Trial Investigating an Integrated Nursing Educational Program to Mitigate Chemotherapy-Induced  Nausea and Vomiting in Cancer Patients: The NIV-EC Trial " aimed  to evaluate the role of a nursing integrated educational approach (i.e., an informative booklet plus an oral educational session) in reducing delayed and breakthrough CINV in cancer Italian patients undergoing to a first-line chemotherapy.

There are many minor issues I'd like the authors resolve.

Abstract

1-Change the keywords. Delete the words "Chemotherapy-induced nausea and vomiting (CINV)", "patient education", "patient empowerment"and "nursing intervention".  Not found in the MeSH (Medical Subject Headings). 

2-The title is appropriate. The authors identify the study design.

Introduction

3-Adequate: The most important concepts of the subject to be developed are identified.

Materials and Methods

4-It is recommended to indicate if hidden allocation of participants was made.

5-The authors indicate the study design, the marked inclusion criteria as well as the procedure they followed.

6-The statistical analysis is adequate.

Results

  • 7-It is recommended to put at the end of each table the abbreviations used. For example, indicate what is meant by "95% CI".
  • 8-It is recommended to add a magnitude of association to assess the strength of association between the significant variables.

Discussion

  • 9-adequate: The discussion is well argued. The authors accept the limitations of the methodology used.

Conclusion

  • 10-Adequate:The objectives are answered in the conclusions.

Reference:

  • 11-adequate: Complies with the journal's standards.

Round 2

Reviewer 3 Report

Comments and Suggestions for Authors

The authors' responses are not satisfactory